# Provable Variational Inference for Constrained Log-Submodular Models

**Josip Djolonga**
Dept. of Computer Science
ETH Zürich
josipd@inf.ethz.ch

**Stefanie Jegelka**
CSAIL
MIT
stefje@csail.mit.edu

**Andreas Krause**
Dept. of Computer Science
ETH Zürich
krausea@ethz.ch

## Abstract

Submodular maximization problems appear in several areas of machine learning and data science, as many useful modelling concepts such as diversity and coverage satisfy this natural diminishing returns property. Because the data defining these functions, as well as the decisions made with the computed solutions, are subject to statistical noise and randomness, it is arguably necessary to go beyond computing a single approximate optimum and quantify its inherent uncertainty. To this end, we define a rich class of probabilistic models associated with constrained submodular maximization problems. These capture log-submodular dependencies of arbitrary order between the variables, but also satisfy hard combinatorial constraints. Namely, the variables are assumed to take on one of — possibly exponentially many — set of states, which form the bases of a matroid. To perform inference in these models we design novel variational inference algorithms, which carefully leverage the combinatorial and probabilistic properties of these objects. In addition to providing completely tractable and well-understood variational approximations, our approach results in the minimization of a convex upper bound on the log-partition function. The bound can be efficiently evaluated using greedy algorithms and optimized using any first-order method. Moreover, for the case of facility location and weighted coverage functions, we prove the first constant factor guarantee in this setting — an efficiently certifiable $e/(e-1)$ approximation of the log-partition function. Finally, we empirically demonstrate the effectiveness of our approach on several instances.

## 1 Introduction

Many real-world tasks can be modeled as distributions over combinatorial objects such as trees, assignments or selections. As an illustrative example, let us consider the following scenario inspired by the recent work of Celis et al. [1]. Assume that we are building a news aggregator and are faced with the task of populating the limited number of slots on the front page with articles originating from various news outlets. We furthermore assume that we have a function that, given a news article and a slot, estimates how good of a match they are. Hence, if we decide that a certain subset of the articles should be shown, we can compute their optimal assignment using a maximal bipartite matching. Furthermore, to make sure that a diverse set of points of views are represented, we want the chosen articles to not only have a high matching value, but to also come from different sources. This can be enforced using a hard selection constraint — for example, we can require that each source $j$ has exactly $k_j$ articles on the front page. While the optimization problem has been well-studied as it is that of submodular maximization, taking a probabilistic approach seems very challenging. Not only the random variables have to satisfy complicated combinatorial requirements, but the utility function is only implicitly defined via optimal matchings and is very challenging for many approximate inference techniques. Nevertheless, by exploiting the submodular properties of the objective and

the combinatorial and probabilistic properties of matroids we will develop a method that can easily handle such models with combinatorial constraints and complex long-ranging variable interactions.

Another family of constraints that often appears is that of (directed) spanning trees. Namely, we are only interested in subsets of edges of a graph that form a spanning tree. Such constraints can model information cascades in network inference [2], or non-projective dependency parse trees in natural language processing [3, 4]. Moreover, the $1$-of-$K$ encoding typically used for multi-label inference tasks is perhaps the simplest and most frequent case of a hard combinatorial assignment constraint.

What all these applications have in common is that they give rise to joint distributions over a set of dependent random variables, each of which is itself a combinatorial object (a spanning tree in network inference and dependency parsing; a discrete selection in multi-label inference and slot allocation). Inference in such combinatorial models is complex due to two sources of dependencies. First, the distribution may express pairwise or higher-order dependencies between elements (in our previous example, the value of the optimal matching). Second, we have strict combinatorial constraints on the support of the distribution (e.g., only trees are allowed) that implicitly induce further interactions.

In this work, we undertake a variational inference approach and approximate these rich distributions with simpler ones that respect the combinatorial constraints but are fully tractable. These approximations posses very strong negative association properties, which we utilize in our theory. To find the optimal approximation we minimize a Rényi divergence over these distributions, which results in efficiently minimizable convex upper bounds on the partition function. While variational inference methods rarely provide any approximation guarantees, our approach yields provably good approximations for certain model families. In summary, this paper makes the following contributions.

- Fast variational convex algorithms for a large family of probabilistic models with submodular dependencies of *arbitrarily high order* in combination with hard combinatorial constraints.

- By combining results from approximate inference and submodular maximization, we prove the first constant factor approximation on the log-partition function for facility location and weighted coverage functions under a family of matroid constraints. We specifically show that our upper bound does not exceed the true value by more than a factor of $(1 - 1/e)$.

- An empirical evaluation of the proposed techniques on several problem instances.

**Related work.** Bouchard-Côté and Jordan [5] introduce a class of variational techniques over combinatorial spaces, but they make a different set of assumptions — they assume a product space and models that are tractable when retaining only one of the constraints. There has also been interest in applying belief propagation (BP) to structured problems such as dependency parsing [3]. Our approach makes a different set of factorization assumptions, and in contrast to BP, provides a bound on the partition function and is guaranteed to converge without any damping heuristics. Other methods that provide upper bounds make factorization assumptions not satisfied by the models we consider [6, 7, 8], or have to repeatedly solve hard optimization problems [9, 10]. MCMC sampling methods for distributions over more general combinatorial objects have been addressed in a rich literature [11]. Li et al. [12] consider distributions over partition and uniform matroids that also allow for non-linear dependencies between the variables and develop Gibbs samplers whose mixing time grows exponentially with the non-linearity of the model. In the unconstrained case, the mixing time as a function of non-submodularity has been analyzed in [13, 14].

Variational inference in *unconstrained* probabilistic submodular models was considered by Djolonga and Krause [15], whose inference method for log-supermodular models was shown to be equivalent to the minimization of the inclusive Rényi divergence [16], which we also use as the variational objective in this paper. The minimization of this divergence for decomposable unconstrained models has been studied in Djolonga et al. [17], who also utilize the $M^\natural$-concavity of the terms. Inference in multi-label log-supermodular models has been considered by Zhang et al. [18]. The tractable distributions used in our variational framework have been already studied [19, 20, 21]. Some of them are determinantal point processes (DPPs), which have been already used in machine learning [22]. Risteski [23] has proved a constant factor approximation for the log-partition function of certain Ising models using a variational approach, and is also leveraging the mean-field bound in the proof.

## 2 Background — submodularity, matroids and continuous extensions

Submodularity [24, 25] formalizes the concept of diminishing returns — the benefit of adding an element decreases with the growth of the context in which it is being included. Formally, a set function $F\colon 2^V \to [0,\infty)$ is said to be submodular if for all $X \subseteq Y \subseteq V$ and $i \in V \setminus Y$ it holds that $F(i \mid X) \geq F(i \mid Y)$, where the *marginal gain* $F(i \mid X)$ is defined as $F(\{i\} \cup X) - F(X)$. To keep the notation as simple as possible , we will w.l.o.g. assume that $V = \{1, 2, \ldots, n\}$.

A classical family of submodular functions are set cover functions. If we associate to each $i \in V$ a set $U_i \subseteq \mathcal{U}$ of elements from some finite universe $\mathcal{U}$, the function is given as the size of the union of the chosen sets, i.e., $F(X) = |\cup_{i \in X} U_i|$. Another well-known function class are facility location functions, defined as $F(X) = \sum_{j=1}^{m} \max_{i \in X} w_{i,j}$ for some non-negative weights $w_{i,j} \geq 0$. The name stems from the following scenario: a set of facilities $V$ serve $m$ customers such that customer $j$ receives a utility of $w_{i,j}$ from facility $i \in V$, and $F(X)$ measures the total utility from the facilities $X$ if each customer can be served by exactly one facility. Moreover, many problems, such as exemplar clustering, which we use in the experimental section, can be modelled using this function class.

**Maximization.** As both examples above model utilities, a natural problem that arises is that of finding a configuration $X \subseteq V$ that maximizes $F$ — cover as much as possible from $\mathcal{U}$, or serve as many customers as possible from the opened facilities. Note that the above functions are not only submodular, but also monotone — adding an item can never decrease the value. Moreover, we typically want to find the maximal $X$ subject to some constraints. A classical problem is that of maximizing over all sets of cardinality at most $k$. In this case, Nemhauser et al. [26] have proven that a simple greedy algorithm results in a provably good solution. Specifically, we start with $X_0 = \emptyset$, and construct the set $X_{j+1}$ as the union of $X_j$ and any element in $\arg\max_{i \in V \setminus X_j} F(i \mid X_j)$. Then, the guarantee is that $F(X_k) \geq (1 - 1/e) \max_{X\colon |X| \leq k} F(X)$, which is also optimal unless P = NP.

$M^\natural$**-concavity.** There exists a subclass of submodular functions for which the above algorithm exactly maximizes the function even when it is not monotone, if we stop once we see a negative gain. These functions, known as $M^\natural$-concave [27, §4], are defined as follows: for all $X, Y \subseteq V$ and $i \in X \setminus Y$ either (i) $F(X) + F(Y) \leq F(X \setminus \{i\}) + F(Y \cup \{i\})$, or (ii) there exist some $j \in Y \setminus X$ such that $F(X) + F(Y) \leq F(X \setminus \{i\} \cup \{j\}) + F(Y \setminus \{j\} \cup \{i\})$. Moreover, it also holds that $X_k = \arg\max_{X\colon |X|=k} F(X)$ [28, Lem. 6.3, 29, 30]. This family contains (see e.g. [31, §3.6]) the maximum function $\max_{i \in X} w_{i,j}$, weighted matroid rank functions, the value of the optimal bipartite matching used in the introduction, as well as functions of the form $F(X) = \sum_{j=1}^{m} \phi_j(|X \cap B_j|)$ for any concave $\phi_j\colon \mathbb{R} \to \mathbb{R}$ and laminar $\{B_j\}_{j=1}^{m}$. While not $M^\natural$-concave themselves, many submodular functions, such as facility location, can be written as sums of $M^\natural$-concave terms — a fact that we will exploit later on in this paper.

**Matroids.** Submodular maximization has been studied not only under cardinality constraints, but also under a broader set of structures that have particularly nice mathematical properties: matroids.

**Definition 1** (Oxley [32]). *A matroid $M$ consists of a ground set $V = \{1, 2, \ldots, n\}$ and a collection $\mathcal{I} \subseteq 2^V$ of subsets of $V$ (called* independent*) that satisfy:*

   *(i) $\emptyset \in \mathcal{I}$.*
   *(ii) If $X \in \mathcal{I}$ and $Y \subseteq X$ then $Y \in \mathcal{I}$.*
   *(iii) If $X \in \mathcal{I}$ and $Y \in \mathcal{I}$ and $|X| < |Y|$ then there exists some $y \in Y \setminus X$ such that $X \cup \{y\} \in \mathcal{I}$.*

A set $X \in \mathcal{I}$ is *maximal* if for all $e \in V \setminus X$, we have that $X \cup \{e\} \notin \mathcal{I}$. We will focus on the case when $\mathcal{M}$ is the collection of all maximal sets in $\mathcal{I}$. These maximal sets are the *bases* of the matroid. This framework encompasses for instance both the cardinality constraints and spanning trees. Namely, the set $\mathcal{I} = \{X \subseteq V \mid |X| \leq k\}$ is known as the uniform matroid and its bases are all subsets of cardinality exactly $k$, while the set of spanning trees form the bases of the graphic matroid, defined as the collection of edge subsets that are cycle-free. This latter example belongs to the family of *regular matroids* that are defined as follows. Let

$$U = [\ \mathbf{u}_1 \mid \mathbf{u}_2 \mid \cdots \mid \mathbf{u}_n\ ] \in \{0, \pm 1\}^{r \times n}$$

be a *totally unimodular* (TU) matrix, meaning that every square submatrix of $U$ has a determinant in $\{0, \pm 1\}$. A subset $X \subseteq V$ is said to be independent if the columns of $U$ indexed by $X$ are linearly

independent. The bases of this matroid are the subsets of the columns of $U$ that form a basis of the column space of $U$. We can think of the $i$-th column $\mathbf{u}_i$ as the *representation* of element $i$. As a concrete example, the graphic matroid of a graph $\mathcal{G} = (\mathcal{V}, \mathcal{E})$ is generated by the (arbitrarily oriented) edge-vertex incidence matrix $U \in \{0, \pm 1\}^{(|\mathcal{V}|-1) \times |\mathcal{E}|}$ of $\mathcal{G}$ after removing an arbitrary vertex.

## 3 The problem and our approach

Formally, we have a random variable $\mathbf{X}$ that takes values in a set of combinatorial objects $\mathcal{M}$. For example, $\mathbf{X}$ could be a random tree drawn from the collection $\mathcal{M}$ of all trees in some graph $\mathcal{G}$. We can think of the members of $\mathcal{M}$ as being the *valid configurations* among all possible sets in $2^V$, so that any configuration not in $\mathcal{M}$ should get a probability of zero. Specifically, we consider distributions over the configurations $\mathcal{M}$ of the general form

$$\mathbb{P}(\mathbf{X} = X) = \frac{[\![X \in \mathcal{M}]\!]}{\mathcal{Z}} \exp\Big(F(X)\Big), \tag{1}$$

where $F \colon 2^V \to \mathbb{R}$ is the *objective function* and $[\![\cdot]\!]$ is the Iverson bracket. Note that the problem of computing the MAP configuration reduces to $\max_{X \in \mathcal{M}} F(X)$, which is can be approximated within a factor of a $(1 - 1/e)$ [33] when $F$ is monotone submodular and $\mathcal{M}$ are matroid bases. We make the following additional assumptions about $F$ and the set $\mathcal{M}$ constituting the support of the distribution.

(i) It holds that $F = \sum_{j=1}^m F_j$ for some monotone $M^\natural$-concave functions $F_j \colon 2^V \to [0, \infty)$.

(ii) The set $\mathcal{M}$ consists of the bases of a matroid, which is a direct sum of uniform and totally unimodular matroids, which we will call *normalizable*.

We would like to point out that the model class is closed under conditioning, as $M^\natural$-concave functions are closed under restrictions, and both uniform and TU matroids are closed under taking minors. The MAP problem under (i) has been studied in [34]. Note that, unlike many inference methods, we make *no assumption* about the number of variables that each $F_j$ depends on, also known as its order.

We will pay special attention to the case when $F$ is a facility location, or equivalently, a weighted coverage, i.e., of the form $F(X) = \sum_{i \in \mathcal{U}} w_i [\![i \in \cup_{i \in X} U_i]\!]$, where $U_i$ and $\mathcal{U}$ are defined as in the unweighted case, and $w_i \geq 0$ are arbitrary weights. As a specific instantiation, let us consider the FLID model of Tschiatschek et al. [35], which has been successfully applied to the problem of item set recommendation. Specifically, we have a set of items $V = \{1, 2, \ldots, n\}$ that we want to recommend to the user. Moreover, we assume that there are a total of $m$ traits, and item $i$ expresses a level of $w_{i,j} \geq 0$ for trait $j \in \{1, 2, \ldots, m\}$. Then, the idea is that the function $F(X) = \sum_{j=1}^m \max_{i \in X} w_{i,j} + \sum_{i \in X} u_i$ captures the classical notion of substitutes — once we select an item that has a high expression level of some item, those items similar to it will be less likely to be included. In addition, there is the modular function $\sum_{i \in X} u_i$ to model the quality of individual items. Similarly to the example in the introduction, we can explicitly enforce the user to see a diverse set of offers by for example presenting them with a fixed number of items from each brand — if items $X_p$ are produced by producer $p$, then we can use $\mathcal{M} = \{X \subseteq V \mid \forall p \colon |X_p \cap X| \leq k_p\}$, also called a partition matroid, which as a direct sum of uniform matroids, satisfies our modelling assumptions.

The central problem of interest in this paper is to compute *marginal probabilities* $\mathbb{P}(Y \subseteq \mathbf{X})$ for any set $Y \subseteq V$. In its general form, this problem is hard, owing to the presence of the intractable normalizer $\mathcal{Z}$, whose computation is also important for the computation of likelihoods and model selection. We therefore revert to approximate techniques for computing the marginal probabilities and the partition function $\mathcal{Z}$. Specifically, we will undertake a divergence minimization approach, which will yield both an estimate of $\log \mathcal{Z}$ and approximate marginals. Namely, we will first define a set of approximate distributions $\mathcal{Q}$ that are rich enough to capture some of the properties of the target distribution $\mathbb{P}$, but are computationally tractable. Then, we will find the distribution $\mathbb{Q}$ in $\mathcal{Q}$ that is the closest to $\mathbb{P}$, as measured by some measure of distributional discrepancy, also called a divergence.

## 4 "Simple" distributions over matroid bases

We begin with a characterization of the distributions $\mathcal{Q}$ that will serve as approximations. These distributions correspond to modular objective functions, so that for some $\boldsymbol{\theta} \in \mathbb{R}^n$ they are given as

$F(X) = \sum_{e \in X} \theta_e = \boldsymbol{\theta}^\top \mathbb{1}_X$, where $\mathbb{1}_X \in \{0,1\}^n$ is the characteristic vector of $X$ with ones only at coordinates $X$. Formally, they belong to the exponential family and can be written as

$$\mathbb{Q}_{\boldsymbol{\theta}}(\mathbf{X} = X) = \exp(\boldsymbol{\theta}^\top \mathbb{1}_X - A(\boldsymbol{\theta}))[\![X \in \mathcal{M}]\!]. \tag{2}$$

Here, $A(\boldsymbol{\theta}) = \log \sum_{X \in \mathcal{C}} \exp(\boldsymbol{\theta}^\top \mathbb{1}_X)$ is the normalizing *log-partition function*, and $\mathcal{M}$ is the set of bases of the considered matroid classes. Because of the constraint $[\![X \in \mathcal{M}]\!]$, the distribution is not a product distribution, and the elements $i \in V$ are not independent. Even though computing $A(\boldsymbol{\theta})$ can be challenging for arbitrary constraints, it can be efficiently done for the considered classes. In what follows we will assume that we have a single normalizable matroid, as the result for their direct sums easily follows. In the uniform matroid case, (2) is known as a cardinality potential, and both $A(\boldsymbol{\theta})$ and the unary marginals can be computed in $O(nk)$ using the algorithm of Tarlow et al. [19]. If $\mathcal{M}$ is a regular matroid, the model can be efficiently normalized via the celebrated matrix-tree theorem.

**Theorem 1** (Maurer [36]). *For regular matroids, it holds that* $A(\boldsymbol{\theta}) = \log \det \sum_{i=1}^n e^{\theta_i} \mathbf{u}_i \mathbf{u}_i^\top$.

Lyons [20] showed that the distribution (2) is a determinantal point process (DPP) with the scaled representation $U_{\boldsymbol{\theta}} = (U \operatorname{diag}(\exp(\theta)) U^\top)^{-1/2} U \operatorname{diag}(\exp(\theta/2))$, and can be marginalized as follows.

**Theorem 2** ([20, Remark 5.6]). *The marginal probability of any* $Y \subseteq V$ *is equal to*

$$P(Y \subseteq \mathbf{X}) = \det K_Y, \tag{3}$$

*where* $K = U_{\boldsymbol{\theta}}^\top U_{\boldsymbol{\theta}} \in \mathbb{R}^{n \times n}$ *and* $K_Y$ *is the submatrix formed by the rows and columns indexed by* $Y$.

For example, the first and second order moments are given by

$$P(e \in X) = \|\mathbf{u}_e^\theta\|^2, \text{ and } P(\{e, e'\} \subseteq X) = \|\mathbf{u}_e^\theta\|^2 \|\mathbf{u}_{e'}^\theta\|^2 - \langle \mathbf{u}_e^\theta, \mathbf{u}_{e'}^\theta \rangle^2, \tag{4}$$

which implies that the elements $e, e'$ are negatively correlated: their joint probability is smaller than if they were independent. Moreover, an even stronger condition can be stated — both cases are strongly Rayleigh [37, Coro. 4.18, Prop. 3.5], so that for any $\mathbb{Q} \in \mathcal{Q}$ we have that $\mathbb{E}_{A \sim \mathbb{Q}}[G(A)H(A)] \leq \mathbb{E}_{A \sim \mathbb{Q}}[G(A)]\mathbb{E}_{A \sim \mathbb{Q}}[H(A)]$ for any monotone functions $G$ and $H$ that depend on disjoint coordinates[1].

As $\mathcal{Q}$ is an exponential family, it has many remarkable properties (for proofs see e.g. [38]), some of which we now present. The marginals, i.e., the vector $\boldsymbol{\mu} \in [0,1]^n$ with entries $\mu_i = \mathbb{E}_{\mathbf{X} \sim \mathcal{Q}_{\boldsymbol{\theta}}}[[\![i \in \mathbf{X}]\!]]$, can be easily computed from the log-partition function as $\boldsymbol{\mu} = \nabla A(\boldsymbol{\theta})$. An important object associated with $\mathcal{Q}$ is the *marginal polytope*, the set of all realizable unary marginals by any distribution over $\mathcal{M}$. In our case, it is equal to the convex hull of the bases, i.e., $\mathbb{M} = \operatorname{conv}\{\mathbb{1}_A \mid A \in \mathcal{M}\}$. Remarkably, $\mathcal{Q}$ is rich enough to represent any marginal vector in $\operatorname{relint} \mathbb{M}$, i.e., $\forall \boldsymbol{\mu} \in \operatorname{relint} \mathbb{M}$ there exists some $\boldsymbol{\theta}(\boldsymbol{\mu}) \in \mathbb{R}^n$ such that $\mathbb{E}_{\mathbf{x} \sim \mathbb{Q}_{\boldsymbol{\theta}(\boldsymbol{\mu})}}[\mathbf{x}] = \boldsymbol{\mu}$. Furthermore, the convex conjugate $A^*(\boldsymbol{\mu})$ of the log-partition function $A$ evaluates to $\infty$ if $\boldsymbol{\mu} \notin \mathbb{M}$, and to $-\mathbb{H}[\mathbb{Q}_{\boldsymbol{\theta}(\boldsymbol{\mu})}]$ otherwise, where $\mathbb{H}[\cdot]$ is Shannon's entropy function. Moreover, we can optimize linear functions over $\mathbb{M}$ using Edmonds' [25] celebrated algorithm in $O(n \log n)$. Namely, to solve $\max_{\boldsymbol{\mu} \in \mathbb{M}} \boldsymbol{\mu}^\top \boldsymbol{\theta}$, first sort $\boldsymbol{\theta}$ in descending order $\theta_{\sigma(1)} \geq \theta_{\sigma(1)} \geq \ldots \geq \theta_{\sigma(n)}$, and define the chain

$$X_0 = \emptyset, \text{ and } X_i = \begin{cases} X_{i-1} \cup \{\sigma(i)\} & \text{if } X_{i-1} \cup \{\sigma(i)\} \in \mathcal{I} \\ X_{i-1} & \text{otherwise} \end{cases}.$$

Then, it can be shown that $\mathbb{1}_{X_n}$ is a maximizer. For spanning trees, this is exactly Kruskal's algorithm.

## 5  Inference using the inclusive infinite Rényi divergence

Having fixed the approximation family, we turn to the choice of the function that will quantify the distance between the distributions, and the analysis of resulting optimization problem. In this paper we will use the *inclusive Rényi $\infty$-divergence* [39, 40], defined as

$$D_\infty(\mathbb{P} \| \mathbb{Q}) = \log \max_{X \in \mathcal{M}} \mathbb{P}(X)/\mathbb{Q}(X). \tag{5}$$

In other words, it evaluates to the worst-case log-ratio between $\mathbb{P}$ and $\mathbb{Q}$. In the terminology of Minka [41], it is an inclusive (zero-avoiding) divergence — it prefers more conservative distributions that do

not assign event probabilities close to zero or one. To better understand the optimization problem that results from minimizing $D_\infty$, let us first define for $F\colon 2^V \to \mathbb{R}$ and any $\mathcal{M} \subseteq 2^V$ the function $\widehat{f}^\star(\mathbf{y} \mid \mathcal{M}) = \inf_{\mathbf{x} \in \mathcal{M}} \mathbf{y}^\top \mathbb{1}_X - F(X)$, which is easily seen to be concave. Then, by expanding the divergence and minimizing with respect to $\mathbb{Q}_{\boldsymbol{\theta}} \in \mathcal{Q}$ we obtain the following upper bound[2]

$$\log \mathcal{Z} \le \inf_{\boldsymbol{\theta} \in \mathbb{R}^n} A(\boldsymbol{\theta}) - \widehat{f}^\star(\boldsymbol{\theta} \mid \mathcal{M}) = \sup_{\boldsymbol{\mu} \in \mathbb{M}} \widehat{f}(\boldsymbol{\mu} \mid \mathcal{M}) - A^*(\boldsymbol{\mu}), \tag{6}$$

where $\widehat{f}(\boldsymbol{\mu} \mid \mathcal{M}) = \widehat{f}^{\star\star}(\boldsymbol{\mu} \mid \mathcal{M}) = \inf_{\mathbf{y} \in \mathbb{R}^n} \boldsymbol{\mu}^\top \mathbf{y} - \widehat{f}(\mathbf{y} \mid \mathcal{M})$ is the concave conjugate of $\widehat{f}^\star(\mathbf{y} \mid \mathcal{M})$, and the equality follows from Fenchel's duality. Unfortunately, we can not evaluate the above bound as we do not know how to compute $\widehat{f}^\star(\boldsymbol{\mu} \mid \mathcal{M})$, which requires the maximization of a non-monotone function over $\mathcal{M}$. We do, however, know that $F$ decomposes as a sum of $M^\natural$-concave functions, which we can leverage to obtain a more tractable bound using dual decomposition [42, 43].

**Proposition 1.** *By applying dual decomposition to* (6) *we arrive at the following bound*

$$\log \mathcal{Z} \le \underbrace{\inf_{\{\boldsymbol{\theta}_j\}_{j=1}^m} A(\sum_{j=1}^m \boldsymbol{\theta}_j) - \sum_{j=1}^m \widehat{f}_j^\star(\boldsymbol{\theta}_j \mid \mathcal{M})}_{\mathcal{R}(\boldsymbol{\theta}_1, \ldots, \boldsymbol{\theta}_m \mid \mathcal{M})} = \sup_{\boldsymbol{\mu} \in \mathcal{M}} \sum_{j=1}^m \widehat{f}_j(\boldsymbol{\mu} \mid \mathcal{M}) - A^*(\boldsymbol{\mu}). \tag{7}$$

Now, instead of maximizing $F$ over $\mathcal{M}$, we only have to maximize only each component $F_j$. Because we have assumed that each $F_j$ is $M^\natural$-concave, if $\mathcal{M}$ is a uniform matroid remember that we can easily solve the resulting problem $\max_{|X|=k} F(X) - \mathbf{y}^\top \mathbb{1}_X$ using the greedy strategy. Even though the general case seems much harder, it can be solved using Murota's duality theorem [27, Thm. 8.21(i)] by introducing a set of $m$ auxiliary variables $\{\boldsymbol{\lambda}_j \in \mathbb{R}^n\}_{j=1}^m$ over which we also have to minimize.

**Proposition 2.** *For any set of parameters* $\{\boldsymbol{\theta}_j \in \mathbb{R}^n\}_{j=1}^m$ *it holds that*

$$A(\sum_{j=1}^m \boldsymbol{\theta}_j) - \sum_{j=1}^m \widehat{f}_j^\star(\boldsymbol{\theta}_j \mid \mathcal{M}) = \inf_{\{\boldsymbol{\lambda}_j\}_{j=1}^m} A(\sum_{j=1}^m \boldsymbol{\theta}_j) - \sum_{j=1}^m \widehat{f}_j^\star(\boldsymbol{\lambda}_j \mid V) + \sum_{j=1}^m \sup_{\boldsymbol{\mu} \in \mathbb{M}} \boldsymbol{\mu}^\top(\boldsymbol{\lambda}_j - \boldsymbol{\theta}_j). \tag{8}$$

Note that it is easy to both evaluate this bound and compute a subgradient. Namely, we can compute both the log-partition function and its derivatives using the methods from Section 4. The computation of both $\widehat{f}^\star$ and the linear maximization over $\mathbb{M}$ can be done using greedy algorithms, and the computed maxima are members of the corresponding subdifferentials. Hence, we can easily employ first-order convex methods to optimize this bound to arbitrary precision in polynomial time.

**The facility location case.** We will now prove a strong theoretical guarantee for the quality of the computed approximation for this important class. Specifically, we will show that the obtained upper bound is no greater than $(1 - 1/e)^{-1} \log \mathcal{Z} \approx 1.582 \log \mathcal{Z}$. To this end, we first construct a lower bound on $\log \mathcal{Z}$, and then show that the lower and upper bounds are within a multiplicative constant of each other. Moreover, this lower bound can be easily evaluated, so that we can at any point return not only a bound, but also a corresponding certificate. We begin by introducing the *multi-linear extension* $\tilde{f}\colon [0,1]^n \to \mathbb{R}$ [33] of $F$, defined as $\tilde{f}(\boldsymbol{\mu}) = \mathbb{E}_{x_i \sim \text{Bernoulli}(\mu_i)}[F(\mathbf{x})]$. It can be evaluated within any accuracy using Monte-Carlo sampling, and also analytically for several cases such as facility location functions (see e.g. [44]). To derive the bound, we start from the mean-field bound [38] (details in appendix) $\mathbb{E}_{X \sim \mathbb{Q}}[F(X)] + \mathbb{H}[\mathbb{Q}] \le \log \mathcal{Z}$, which holds for any distribution $\mathbb{Q}$ absolutely continuous with respect to $\mathbb{P}$. Then, we use a result by Chekuri et al. [45, Lem. VI.1], which states that if $F$ is a weighted sum of coverage functions and $\mathbb{Q}$ is negatively associated with unary marginals $\boldsymbol{\mu} \in [0,1]^n$ — both conditions satisfied for our model — then $\mathbb{E}_{X \sim \mathbb{Q}}[F(X)] \ge \tilde{f}(\boldsymbol{\mu})$.

**Proposition 3.** *If $F$ is a facility location function, then for any $\boldsymbol{\theta} \in \mathbb{R}^n$ it holds that*

$$\mathcal{L}(\boldsymbol{\theta}) = \tilde{f}(\nabla A(\boldsymbol{\theta})) + \mathbb{H}[\mathbb{Q}_{\boldsymbol{\theta}}] = \tilde{f}(\nabla A(\boldsymbol{\theta})) + A(\boldsymbol{\theta}) - \nabla A(\boldsymbol{\theta})^\top \boldsymbol{\theta} \le \log \mathcal{Z}. \tag{9}$$

We will actually prove a stronger result that holds not only for (7), but also if we relax the bound and replace $\widehat{f}_j^\star(\mathbf{y} \mid \mathcal{M})$ by $\widehat{f}_j^\star(\mathbf{y} \mid V)$, i.e., we ignore the constraints when we maximize. In other words,

we will show that the bound

$$\underbrace{\inf_{\{\boldsymbol{\theta}_j\}_{j=1}^m} A(\sum_{j=1}^m \boldsymbol{\theta}_j) - \sum_{j=1}^m f_j^\star(\boldsymbol{\theta}_j)}_{\mathcal{R}(\boldsymbol{\theta}_1,\dots,\boldsymbol{\theta}_j)} = \sup_{\boldsymbol{\mu}\in\mathcal{M}} \sum_{j=1}^m \widehat{f}_j(\boldsymbol{\mu}) - A^*(\boldsymbol{\mu}), \qquad (10)$$

is within a multiplicative constant of $\mathcal{L}$ evaluated at any optimizer of (10). Even though perhaps not immediately clear from their definitions, both $\widehat{f}(\boldsymbol{\mu} \mid V)$ and $\tilde{f}(\boldsymbol{\mu})$ are extensions of $F$ — if we see $F$ as being defined over $\{0,1\}^n$ instead of $2^V$ using the natural bijection, then both of them agree with $F$ for binary vectors and continuously fill in the rest of the unit cube. Moreover, they are closely related via the following result known as the *correlation gap inequality*.

**Theorem 3** ([46, Lem. 3.8, 47]). *If $F\colon 2^V \to \mathbb{R}$ is monotone submodular with $F(\emptyset) = 0$, then*

$$\forall \boldsymbol{\mu} \in [0,1]^n\colon (1-1/e)\widehat{f}(\boldsymbol{\mu} \mid V) \leq \tilde{f}(\boldsymbol{\mu}) \leq \widehat{f}(\boldsymbol{\mu} \mid V).$$

By combining these two results, we can finally prove the approximation result claimed above.

**Theorem 4.** *If $F$ is a facility location function and $\{\boldsymbol{\theta}_j^*\}_{j=1}^m$ minimizes (7) or (10), then*

$$\mathcal{L}(\sum_{j=1}^m \boldsymbol{\theta}_j^*) \leq \log \mathcal{Z} \leq \mathcal{R}(\boldsymbol{\theta}_1^*,\dots,\boldsymbol{\theta}_m^*) \leq (1-1/e)^{-1}\mathcal{L}(\sum_{j=1}^m \boldsymbol{\theta}_j^*) \leq (1-1/e)^{-1}\log \mathcal{Z}. \qquad (11)$$

Furthermore, at any point during the optimization we can easily certify our approximation quality by computing $\mathcal{C}(\boldsymbol{\theta}_1,\dots,\boldsymbol{\theta}_m) = \mathcal{R}(\boldsymbol{\theta}_1,\dots,\boldsymbol{\theta}_m)/\mathcal{L}(\sum_{j=1}^m \boldsymbol{\theta}_j^*)$, as the true approximation factor $\mathcal{R}(\boldsymbol{\theta}_1,\dots,\boldsymbol{\theta}_m)/\log \mathcal{Z}$ is guaranteed to be upper bounded by it.

## 6 Experiments

We perform numerical experiments to better understand the practical performance of the proposed methods, namely how good is the approximation when compared to the theoretical $e/(e-1)$ factor and how well are the marginals estimated. Moreover, we showcase the scalability of our approach by performing inference on large real-world instances. The implementation was done in Python using PyTorch, and we optimize the bound using subgradient descent. The computation of the log-partition function and its gradients (building on the code from [48]), as well as the greedy oracle were implemented in C++. We provide all details in the appendix.

### 6.1 Synthetic experiments

We begin by comparing the accuracy of the methods on a set of synthetic experiments. We consider facility location models with objectives of the form $F(X) = \sum_{j=1}^{20} \max_{i\in X} w_{i,j}$, where we sample $w_{i,j} \sim \text{Uniform}[0,\alpha]$. We vary the inverse temperature parameter $\alpha$ and show the results in Figure 1. We first used a uniform matroid constraint $|X| = 5$ over a ground set of size $n = 40$. For the same models we then considered partition constraints by partitioning $V$ into three sets $V_1$, $V_2$ and $V_3$ of sizes 10, 10, and 20 respectively and defining $\mathcal{M} = \{X \subseteq V \mid |X\cap V_1| = 2, |X\cap V_2| = 2, |X\cap V_3| = 4\}$. Because the number of configurations is in the millions, we were able to compute the exact marginals and log-partition functions. From the plots we can see that the approximation is much better than the theoretical factor ($\approx 1.582$), and close to exact in the small and high temperature regimes. Moreover, even though the divergence we are optimizing does not necessarily target the marginals, we can see that they are also approximated within a small error.

### 6.2 Real data

We consider two problems from data mining that can be written as facility location maximization problems under cardinality constraints. For each function $F(A)$ we perform inference in models with objectives $\alpha F(A)$ for varying $\alpha \geq 0$. Moreover, to obtain statistical estimates on the approximation factors, we repeat the experiments several times by taking random subsets of the data.

*Exemplar clustering.* Given a dataset $\mathcal{X} = \{\mathbf{x}_1, \mathbf{x}_2, \dots, \mathbf{x}_n\}$ of $n$ points in $\mathbb{R}^d$, we want to find a small subset of size $k = 10$ that is a good summary of $\mathcal{X}$ by minimizing $G(A) = \sum_{i=1}^n \min_{\mathbf{x}_j\in A} \|\mathbf{x}_i - \mathbf{x}_j\|$.

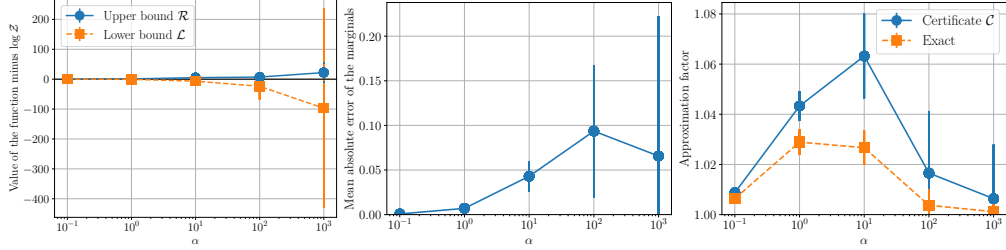

(a) Inference on synthetic models under a uniform matroid constraint $|X| = 6$.

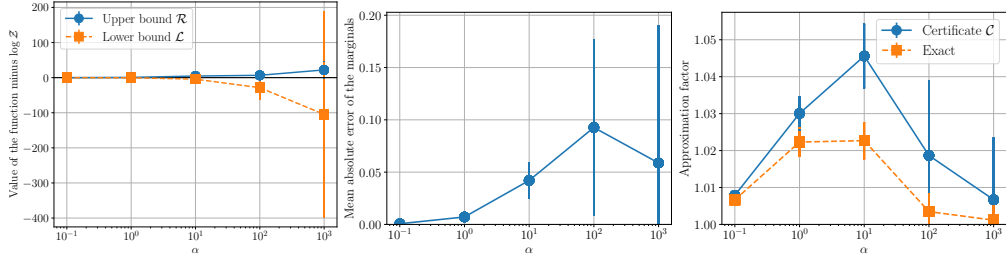

(b) Inference on synthetic models under a partition matroid constraint with 3 blocks of sizes 10, 10 and 15.

Figure 1: Results on synthetic facility location models on a ground set of size $n = 40$. The parameters are sampled from UNIFORM$(0, \alpha)$, and there are $m = 10$ components. The ordinates on plots in the first column have been centered so that zero corresponds to the true partition function. In the last column we plot both the certified approximation factor (the ratio of the upper bound and the certificate) and the exact one (when dividing by the exact partition function). The error bars indicate three standard deviations from 20 repetitions.

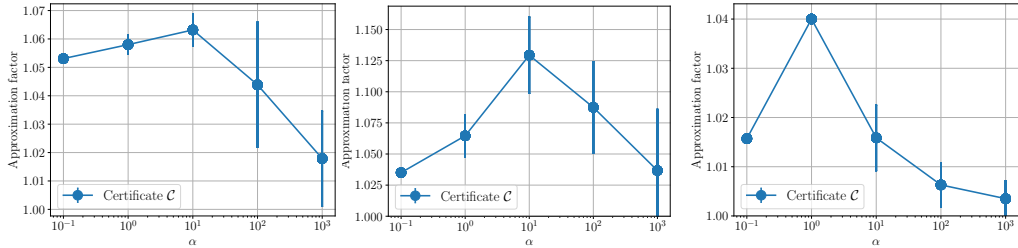

(a) Sensor placement under uniform (left) and partition (right) matroids. (b) Exemplar clustering (CIFAR10).

Figure 2: Results on large real-world datasets (full explanation in §6.2). The error bars indicate three standard deviations from 20 repetitions. Note that the certificate is significantly lower than the theoretical factor of 1.582.

While $-G$ is not submodular, it can be shown [49] that $F(A) = G(\{\mathbf{x}_0\}) - G(A \cup \{\mathbf{x}_0\})$ is monotone submodular for carefully chosen $\mathbf{x}_0$, typically taken to be the origin. We show our results in Figure 2(b), on $n = 1500$ points from the CIFAR10 [50] dataset normalized as in [44].

*Sensor placement.* The second problem is that of placing sensors at pipe junctions in order to effectively detect water contaminations. Namely, there a total of $n$ locations where we can place our sensors, and a set of $m$ possible contamination scenarios. For each scenario $j$ and sensor $i$ there is some utility $w_{i,j} \geq 0$ if $i$ detects contamination $j$, computed e.g. as a function of the the detection time, and the total utility is naturally captured using $F(A) = \sum_{j=1}^{m} \max_{i \in A} w_{i,j}$. We use a subset of the data from [51], and show the results in Figure 2(a). We consider two scenarios — (i) $n = 5000, m = 300$ under a cardinality constraint $\mathcal{M} = \{X \subseteq V \mid |X| = 50\}$, and (ii) $n = 1500, m = 100$ under a partition matroid, constructed by splitting $V$ into 3 blocks of equal size, and consider only distributions that pick exactly 5, 10 and 5 points from each block respectively.

Despite the fact that these models have a much larger number of variables and components in the objective, in Figure 2 we see a behaviour similar to that of the synthetic instances — the certificate of the approximation factor is close to one under high and low temperatures (large and small $\alpha$ respectively), while remaining always significantly smaller than the theoretical guarantee.

# 7 Conclusion

We explored a new, rich class of probabilistic models, whose variables realize bases of a sum of normalizable matroids. These models allow to capture high-order submodular dependencies between complex combinatorial objects. We presented efficient, convergent convex variational inference algorithms that yield upper bounds on the partition function. Moreover, we proved the first constant factor approximation on the log-partition function of facility location and weighted models under constraints. We also numerically showcased the quality of the estimated partition function and the marginals. Our models and methods provide important steps towards exploiting combinatorial structure for principled modeling and reasoning about complex real-world phenomena.

**Acknowledgements.** The research was partially supported by ERC StG 307036, Google European PhD Fellowship, and NSF CAREER award 1553284.

## Footnotes

[1]The case $\mathcal{M} = \{X \mid |X| \leq k\}$ can be also normalized and $\mathbb{Q}$ is again strongly Rayleigh [37, Cor. 4.18].

[2]We defer the proofs of all results in this section to the appendix.

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
