[Supplementary Material]

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

# A Proofs

To simplify the notation we use $\widehat{f}(\boldsymbol{\mu}) = \widehat{f}(\boldsymbol{\mu} \mid V)$ and $\widehat{f}^*(\mathbf{y} \mid V) = \widehat{f}^*(\mathbf{y})$, and also treat the set functions as being defined over the boolean lattice.

**Lemma 1.** *From the inclusive infinite Rényi divergence we have the following bound*

$$\log \mathcal{Z} \le A(\boldsymbol{\theta}) - \widehat{f}^\star(\boldsymbol{\theta} \mid \mathcal{M}).$$

*Proof.*

$$
\begin{aligned}
0 \le D_\infty(\mathbb{P} \,\|\, \mathbb{Q}_{\boldsymbol{\theta}}) &= \log \sup_{X \in \mathcal{M}} \mathbb{P}(A)/\mathbb{Q}_{\boldsymbol{\theta}}(A) \\
&= \log \sup_{X \in \mathcal{M}} \exp(F(X) - \log \mathcal{Z})/\exp(\boldsymbol{\theta}^\top \mathbb{1}_X - A(\boldsymbol{\theta})) \\
&= \sup_{X \in \mathcal{M}} \log \exp(F(X) - \log \mathcal{Z})/\exp(\boldsymbol{\theta}^\top \mathbb{1}_X - A(\boldsymbol{\theta})) \\
&= \sup_{X \in \mathcal{M}} F(X) - \log \mathcal{Z} - \boldsymbol{\theta}^\top \mathbb{1}_X + A(\boldsymbol{\theta})) \\
&= A(\boldsymbol{\theta}) - \log \mathcal{Z} + \sup_{X \in \mathcal{M}} F(X) - \boldsymbol{\theta}^\top \mathbb{1}_X \\
&= A(\boldsymbol{\theta}) - \log \mathcal{Z} - \inf X \in \mathcal{M} \boldsymbol{\theta}^\top \mathbb{1}_X - F(X) \\
&= A(\boldsymbol{\theta}) - \log \mathcal{Z} - \widehat{f}^\star(\boldsymbol{\theta} \mid \mathcal{M}).
\end{aligned}
$$

$\square$

*Proof of Proposition 1.*

$$
\begin{aligned}
\log \mathcal{Z} &\le A(\boldsymbol{\theta}) - f^*(\boldsymbol{\theta} \mid \mathcal{M}) \\
&= A(\boldsymbol{\theta}) - \inf_{\mathbf{x} \in \mathcal{M}} \mathbf{x}^\top \boldsymbol{\theta} - \sum_{j=1}^m F_j(\mathbf{x}) \\
&= A(\boldsymbol{\theta}) - \inf_{\substack{\mathbf{x}_j \in \mathcal{M}, \mathbf{x} \in \mathbb{R}^n \\ \text{s.t.} \mathbf{x} = \mathbf{x}_j}} \mathbf{x}^\top \boldsymbol{\theta} - \sum_{j=1}^m F_j(\mathbf{x}_j) \\
&= A(\boldsymbol{\theta}) - \inf_{\mathbf{x}_j \in \mathcal{M}} \inf_{\mathbf{x} \in \mathbb{R}^n} \sup_{\boldsymbol{\theta}_j} \sum_{j=1}^m \boldsymbol{\theta}_j^\top (\mathbf{x} - \mathbf{x}_j) + \mathbf{x}^\top \boldsymbol{\theta} - \sum_{j=1}^m F_j(\mathbf{x}_j) \\
&\le A(\boldsymbol{\theta}) - \sup_{\boldsymbol{\theta}_j} \inf_{\mathbf{x}_j \in \mathcal{M}} \inf_{\mathbf{x} \in \mathbb{R}^n} \sum_{j=1}^m \boldsymbol{\theta}_j^\top (\mathbf{x}_j - \mathbf{x}) + \mathbf{x}^\top \boldsymbol{\theta} - \sum_{j=1}^m F_j(\mathbf{x}_j) \\
&= A(\boldsymbol{\theta}) - \sup_{\boldsymbol{\theta}_j} \inf_{\substack{\mathbf{x}_j \in \mathcal{M} \\ \text{s.t.} \sum_j \boldsymbol{\theta}_j = \boldsymbol{\theta}}} \sum_{j=1}^m \boldsymbol{\theta}_j^\top \mathbf{x}_j - \sum_{j=1}^m F_j(\mathbf{x}_j) \\
&= \inf_{\boldsymbol{\theta}_j} A\left(\sum_{j=1}^m \boldsymbol{\theta}_j\right) + \sum_{j=1}^m \widehat{f}_j^\star(\boldsymbol{\theta} \mid \mathbb{M}).
\end{aligned}
$$

The equality is immediately applied from Fenchel's theorem, which holds because $A$ is continuous everywhere. $\square$

*Proof of Proposition 2.* Note that

$$\widehat{f}^\star(\boldsymbol{\theta} \mid \mathcal{M}) = \inf_{\mathbf{x} \in \mathcal{M}} \mathbf{y}^\top \mathbf{x} - F(\mathbf{x}) = \inf_{\mathbf{x} \in \mathbb{R}^n} I_{\mathcal{M}}(\mathbf{x}) + \mathbf{y}^\top \mathbf{x} - F(\mathbf{x}),$$

where $I_{\mathcal{M}}$ is the indicator on $\mathcal{M}$, i.e, evaluating to 0 on $\mathcal{M}$ and $\infty$ elsewhere. Because of the matroid basis exchange property, we have that $I_{\mathcal{M}}$ is $M^\natural$-convex as $-I_{\mathcal{M}}$ satisfies condition (ii) of

the definition. This implies that $I_\mathcal{M}(\mathbf{x}) + \mathbf{y}^\top \mathbf{x}$ is also $M^\natural$-convex, so by applying [27, Theorem 8.21(i)] we have that

$$\inf_{\mathbf{x} \in \mathbb{R}^n} I_\mathcal{M}(\mathbf{x}) + \boldsymbol{\theta}^\top \mathbf{x} - F(\mathbf{x}) = \sup_{\boldsymbol{\lambda} \in \mathbb{R}^n} \widehat{f}^\star(\boldsymbol{\lambda}) - (I_\mathcal{M}(\mathbf{x}) + \boldsymbol{\theta}^\top \mathbf{x})^*(\boldsymbol{\lambda}).$$

Now, because $(I_\mathcal{M}(\mathbf{x}) + \boldsymbol{\theta}^\top \mathbf{x})^*(\boldsymbol{\lambda}) = I_\mathcal{M}^*(\boldsymbol{\lambda} - \boldsymbol{\theta})$, we finally arrive at

$$-\widehat{f}^\star(\boldsymbol{\theta} \mid \mathcal{M}) = -\inf_{\mathbf{x} \in \mathbb{R}^n} I_\mathcal{M}(\mathbf{x}) + \boldsymbol{\theta}^\top \mathbf{x} - F(\mathbf{x}) = -\sup_{\boldsymbol{\lambda} \in \mathbb{R}^n} \widehat{f}^\star(\boldsymbol{\lambda}) - I_\mathcal{M}^*(\boldsymbol{\lambda} - \boldsymbol{\theta}) = \inf_{\boldsymbol{\lambda} \in \mathbb{R}^n} I_\mathcal{M}^*(\boldsymbol{\lambda} - \boldsymbol{\theta}) - \widehat{f}^\star(\boldsymbol{\lambda}),$$

which is exactly what we had to show as $I_\mathcal{M}^*(\mathbf{z}) = \sup_{X \in \mathcal{M}} \mathbf{z}^\top \mathbb{1}_X = \sup_{\boldsymbol{\mu} \in \mathbb{M}} \mathbf{z}^\top \boldsymbol{\mu}$.   $\square$

*Proof of Proposition 3.* Because each $\mathbb{Q} \in \mathcal{Q}$ has the same support as $\mathbb{P}$, by expanding the KL divergence we obtain the following classic bound

$$0 \leq D_{\mathrm{KL}}(\mathbb{Q} \,\|\, \mathbb{P}) = \sum_{A \in \mathcal{M}} \mathbb{Q}(A) \log[\mathbb{Q}(A)/\mathbb{P}(A)] = \log \mathcal{Z} - \mathbb{E}_{A \sim \mathbb{Q}}[F(A)] - \mathbb{H}[\mathbb{Q}]. \tag{12}$$

If we re-arrange the terms we can see that minimizing the KL divergence is equivalent to the maximization of $\mathbb{E}_{A \sim \mathbb{Q}_{\boldsymbol{\theta}}}[F(A)] + \mathbb{H}[\mathbb{Q}_{\boldsymbol{\theta}}] \leq \log \mathcal{Z}$. Then, because the variables are strongly Rayleigh under $\mathbb{Q}$, we can apply Chekuri et al. [45, Lemma VI.1], which implies that $\mathbb{E}_{A \sim \mathbb{Q}_{\boldsymbol{\theta}}}[F(A)] \geq \tilde{f}(\nabla A(\boldsymbol{\theta}))$. Hence, it is indeed true that $\mathcal{L}(\boldsymbol{\theta}) = \tilde{f}(\nabla A(\boldsymbol{\theta})) + \mathbb{H}[\mathbb{Q}_{\boldsymbol{\theta}}] \leq \mathbb{E}_{A \sim \mathbb{Q}_{\boldsymbol{\theta}}}[F(A)] \leq \log \mathcal{Z}$. We would like to note that Chekuri et al. [45, Lemma VI.1] is stated for coverage functions and easily extended for weighted coverages, but also holds for facility locations as they can be written as sums of weighted coverages (see e.g. [44, Lemma 5]). Finally, the relationship $-A^*(\nabla A(\boldsymbol{\theta})) = A(\boldsymbol{\theta})$ is a direct result of the classical relationship between $\boldsymbol{\theta}$ and $\boldsymbol{\mu}$, but can be also easily seen directly as

$$\mathbb{H}[\mathbb{Q}_{\boldsymbol{\theta}}] = -\mathbb{E}_{\mathbb{Q}_{\boldsymbol{\theta}}}[\log \mathbb{Q}_{\boldsymbol{\theta}}(\mathbf{x})] = -\mathbb{E}_{\mathbb{Q}_{\boldsymbol{\theta}}}[\boldsymbol{\theta}^\top \mathbb{1}_X - A(\boldsymbol{\theta})] = A(\boldsymbol{\theta}) - \mathbb{E}_{\mathbb{Q}_{\boldsymbol{\theta}}}[\boldsymbol{\theta}^\top \mathbb{1}_X] = A(\boldsymbol{\theta}) - \boldsymbol{\theta}^\top \nabla A(\boldsymbol{\theta}).$$

$\square$

*Proof of Theorem 4.* Let $\boldsymbol{\mu}^* \in \mathbb{M}$ be the variable that achieves the maximum, and let $\boldsymbol{\theta}^* = \sum_{j=1}^m \boldsymbol{\theta}_j^*$ be the optimal primal parameters, so that $\boldsymbol{\mu}^*$ holds the marginals of $\mathbb{Q}_{\boldsymbol{\theta}^*}$.

Note that for each $j \in \{1, 2, \ldots, m\}$ it holds that $\widehat{f}_j^*(\boldsymbol{\mu} \mid \mathcal{M}) \geq \widehat{f}_j^*(\boldsymbol{\mu})$ as the latter is defined over a strictly larger set, which implies the reverse relationship between the conjugates $\widehat{f}_j(\boldsymbol{\mu} \mid \mathcal{M}) \leq \widehat{f}_j(\boldsymbol{\mu})$.

$$\begin{aligned}
\log \mathcal{Z} &\leq A\Big(\sum_{j=1}^m \boldsymbol{\theta}_j^*\Big) - \sum_{j=1}^m f_j^\star(\boldsymbol{\theta}_j \mid \mathcal{M}) \\
&= \sum_{j=}^m f_j(\boldsymbol{\mu}^* \mid \mathcal{M}) - A^*(\boldsymbol{\mu}^*) && \text{(Fenchel duality)} \\
&\leq \sum_{j=}^m \widehat{f}_j(\boldsymbol{\mu}^*) - A^*(\boldsymbol{\mu}^*) && \text{(Argument above)} \\
&\leq (1 - 1/e)^{-1} \sum_{j=1}^m \tilde{f}_j(\boldsymbol{\mu}^*) - A^*(\boldsymbol{\mu}^*) && \text{(Theorem 3)} \\
&\leq (1 - 1/e)^{-1}\Big(\sum_{j=}^m \tilde{f}_j(\boldsymbol{\mu}^*) - A^*(\boldsymbol{\mu}^*)\Big) && (-A^* = \mathbb{H} \geq 0) \\
&= (1 - 1/e)^{-1} \mathcal{L}(\boldsymbol{\theta}^*) \\
&\leq (1 - 1/e)^{-1} \log \mathcal{Z} && \text{(Proposition 3)}.
\end{aligned}$$

$\square$

# B   Experimental details

Before multiplying by $\alpha$ the facility location functions were normalized so that $\max_{i,j} w_{i,j} = 1$.

| Figure 1(a) | $\alpha$ | .1 | 1 | 10 | 100 | 1000 |
|---|---|---|---|---|---|---|
|  | $\gamma$ | .1 | .1 | .5 | 2 | 4 |
|  | $\theta$ | 1 | 1 | 1 | 1 | 1 |
|  | $t$ | 3000 | 3000 | 3000 | 3000 | 3000 |
| Figure 1(b) | $\alpha$ | .1 | 1 | 10 | 100 | 1000 |
|  | $\gamma$ | .1 | .1 | .5 | 2 | 4 |
|  | $\theta$ | 1 | 1 | 1 | 1 | 1 |
|  | $\lambda$ | 1 | 1 | 1 | 1 | 1 |
|  | $t$ | 3000 | 3000 | 3000 | 3000 | 3000 |
| Figure 2(a) (left) | $\alpha$ | .1 | 1 | 10 | 100 | 1000 |
|  | $\gamma$ | 1 | 1 | 1 | 1 | 1 |
|  | $\theta$ | 1 | 1 | 1 | 1 | 1 |
|  | $t$ | 4000 | 4000 | 4000 | 4000 | 4000 |
| Figure 2(a) (right) | $\alpha$ | .1 | 1 | 10 | 100 | 1000 |
|  | $\gamma$ | .5 | .5 | .5 | .5 | 2 |
|  | $\theta$ | 1 | 1 | 1 | 1 | 1 |
|  | $\lambda$ | 1 | 1 | 1 | 1 | 1 |
|  | $t$ | 2000 | 2000 | 2000 | 2000 | 2000 |
| Figure 2(b) | $\alpha$ | .1 | 1 | 10 | 100 | 1000 |
|  | $\gamma$ | .01 | 1 | 2 | 4 | 8 |
|  | $\theta$ | .01 | 1 | 1 | 1 | 1 |
|  | $t$ | 2000 | 2000 | 2000 | 2000 | 2000 |

Table 1: We used gradient descent with an initial rate of $\gamma$ and halved it every tenth of the number of iterations $t$. We initialized the parameters as $\boldsymbol{\theta}_j \sim \text{Uniform}(0, \theta)$ and $\boldsymbol{\lambda}_j \sim \text{Uniform}(0, \gamma)$.