[Reviews · NeurIPS 2018]

Reviewer 1



This article provides a theoretical study of statistical inference over some discrete optimization models, namely the maximization of submodular problems with a cardinality constraints, encompassing a class of discrete probabilistic models. The paper is sound, well written and pleasant to read. The supplementary material is minimal but useful. The authors claim their class is new and "rich" but fail to make a convincing case of the latter statement, instead revisiting fairly classical problems in their experimental section. Their first studied problem is monotone submodular, and so is relatively simple to optimize, and their second is a partition matroid, also simple. I'm also wondering if NIPS is the correct venue for such an article. The connection to machine learning is somewhat tenuous, I believe the authors could have made a better effort for relevance in their experimental section.

Reviewer 2



The authors present an algorithm for approximate inference in exponential family models over the bases of a given matroid. In particular, the authors show how to leverage standard variational methods to yield a provable approximation to the log partition function of certain restricted families. This is of interest as 1) these families can be difficult to handle using standard probabilisitic modeling approaches and 2) previous bounds derived from variational methods do not necessarily come with performance guarantees. General comments: Although the approach herein would be considered variational approximation methods, the term variational inference is more commonly used for a related but different optimization problem. There are lots of other variational approaches that yield provable upper/lower bounds on the partition function. Some effort should be made to cite these. A few to help the authors get started (see citations therein for more): Tamir Hazan and Amnon Shashua. "Norm-product belief propagation: Primal-dual message-passing for approximate inference." IEEE Transactions on Information Theory 56, no. 12 (2010): 6294-6316. Meltzer, T., Globerson, A. and Weiss, Y., 2009, June. Convergent message passing algorithms: a unifying view. In Proceedings of the twenty-fifth conference on uncertainty in artificial intelligence (pp. 393-401). AUAI Press. Nicholas Ruozzi. Beyond log-supermodularity: lower bounds and the Bethe partition function. Uncertainty in Artificial Intelligence (UAI), July 2013. Nicholas Ruozzi. The Bethe partition function of log-supermodular graphical models. Advances in Neural Information Processing Systems (NIPS), December 2012. Typos: Line 222, "which is easily seen to concave" -> "which is easily seen to be concave"

Reviewer 3



In this paper, the authors study the problem of constrained log-submodular models and propose new variational inference algorithms. Overall, this is a good paper. Here are some detailed comments 1. I notice that this paper focus on facility location and weighted coverage functions and uses gradient methods, which is very similar to the analyses in [39] Karimi and Lucic 2017. Both papers get the same bound of e/(e-1). The authors have cited [39] but did not give enough comparison or discussion. Also, if the bound holds for weighted coverage functions, it is not necessary to discuss facility locations since the latter is a subclass of the former. 2. Is the theoretical upper bound e/(e-1) tight? 3. line 122- 125, in the definition of matroids, one statement is missing, i.e. \emptyset in I. 4. Line 63 has double “that”